# Novel Insights into Circular RNAs in Metastasis in Breast Cancer: An Update

**DOI:** 10.3390/ncrna9050055

**Published:** 2023-09-16

**Authors:** Paola Zepeda-Enríquez, Macrina B. Silva-Cázares, César López-Camarillo

**Affiliations:** 1Posgrado en Ciencias Genómicas, Universidad Autónoma de la Ciudad de México, CDMX 03100, Mexico; paola.zepeda@estudiante.uacm.edu.mx; 2Coordinación Academica Región Altiplano, Universidad Autónoma de San Luis Potosí, Matehuala 78700, Mexico; macrina.silva@uaslp.mx

**Keywords:** breast cancer, circular RNAs, metastasis, exosomes, therapy resistance

## Abstract

Circular RNAs (circRNAs) are single-stranded closed non-coding RNA molecules that are aberrantly expressed and produce tumor-specific gene signatures in human cancers. They exert biological functions by acting as transcriptional regulators, microRNA sponges, and protein scaffolds, regulating the formation of protein–RNA complexes and, ultimately, regulating gene expression. Triple-negative breast cancer (TNBC) is one of the most aggressive cancers of the mammary gland and has a poor prognosis. Studies of circRNAs in TNBC are limited but have demonstrated these molecules’ pivotal roles in cell proliferation, invasion, metastasis, and resistance to chemo/radiotherapy, suggesting that they could be potential prognostic biomarkers and novel therapeutic targets. Here, we reviewed the status of actual knowledge about circRNA biogenesis and functions and summarized novel findings regarding their roles in TNBC development and progression. In addition, we discussed recent data about the importance of exosomes in the transport and export of circRNAs in TNBC. Deep knowledge of circRNA functions in metastasis and therapy responses could be an invaluable guide in the identification of novel therapeutic targets for advancing the treatment of TNBC.

## 1. Introduction

The Global Cancer Observatory (GLOBOCAN 2020) has reported an increase in breast cancer incidence rates worldwide, as well as an estimated 2,261,419 million new cases of breast cancer and 684,996 breast cancer deaths [1]. The foregoing is associated with the human development index, and the higher incidence rates were recorded for countries that are in transitionary states. The risk factors for breast cancer development include advanced age, as well as hormonal factors such as an early or late age of menarche, fewer pregnancies and, therefore, less infant breastfeeding, and the use of contraceptives or hormonal therapies during menopause. Environmental risk factors include alcohol intake, obesity, a sedentary lifestyle, and smoking [2]. Due to its molecular heterogeneity, breast cancer has been classified according to the expressions of the estrogen receptor (ER), the progesterone receptor (PR), and human epidermal growth factor receptor 2 (HER2), resulting in three subtypes: luminal A and B, HER2-enriched, and triple-negative or basal-like breast cancer [3,4,5]. TNBC is a very heterogeneous subtype of cancer that has the worst prognosis of all cancers due to a lack of therapeutic targets and a resistance to chemotherapy. Therefore, the options for its treatment are limited. The treatment of breast cancer depends on its stage; the early stages have the best prognoses with breast-conserving surgery and radiotherapy, whereas at later stages, cytotoxic chemotherapy is used in addition to mastectomy [6,7]. TNBC has shown expressions of the epidermal growth factor receptor (EGFR), CK5/6, CK14, and CK17, which has cataloged it as a basal subtype. Recently, TNBC was classified into five subtypes, including immunomodulatory (IM), luminal androgen receptor (LAR), basal-like 1 (BL-1), basal-like 2 (BL-2), mesenchymal (M), and mesenchymal stem-like (MSL); each has its own molecular characteristics that may be the keys to designing new effective therapies against TNBC [8,9,10,11]. The diagnosis of triple-negative breast cancer is made mainly with mammogram or ultrasound imaging of the breast combined with nuclear magnetic resonance. The use of liquid biopsy is a promising diagnostic method to detect the presence of circulating tumor cells, tumor-derived extracellular vesicles (exosomes), circulating tumor nucleic acids, and oncogenic microRNAs [12,13,14]. Traditional cancer treatment is based on combinations of taxanes, anthracyclines, cyclophosphamide, cisplatin, and fluorouracil in the early stages of the disease, whereas in advanced stages, the gold standard is neoadjuvant chemotherapy [15]. Immunotherapy involves the use of CTLA4 and PD-L1 inhibitors (nivolumab and pembrolizumab), since both molecules prevent the activation of the adaptive immune response in cancer and have exhibited good clinical responses [16,17]. However, the effectiveness of these treatments remains limited mainly because of three challenging factors: (i) late diagnosis in the advanced stages of breast cancer, (ii) development of resistance to cancer chemotherapy, and (iii) an apparition of metastasis to distant tissues, which frequently accompanies the developed resistance and jeopardizes the survival of cancer patients.

CircRNAs are a novel class of closed endogenous non-coding RNAs that are involved in multiple physiological process in eukaryotic cells. Their functions are associated with the regulation of gene expression though diverse processes, including acting as (i) microRNA sponges, (ii) scaffolds of proteins (RNA-binding proteins), and (iii) modulators of mRNA synthesis, (iv) splicing, and (v) protein production. Increasing evidence has indicated that circRNAs may play prominent roles in the development and progression of human cancers. In this review, we discussed the roles of circRNAs in triple-negative breast cancer and focused on their functions in metastasis and resistance to chemotherapy, highlighting their potential uses as biomarkers and therapeutic targets [18].

## 2. Metastasis in Breast Cancer: A Challenging Hallmark

Metastasis is the most devastating feature of malignant neoplasia and is responsible for a large number of cancer-related deaths. Disseminated disease occurs when highly aggressive cancer cells detach from a primary tumor, migrating to distant tissues and organs and achieving complete adaptation to a new niche to initiate the growth of a secondary malignant tumor. Breast cancer metastasis occurs mainly in the bones, lungs, liver, and brain, which drastically shortens the overall survival of patients [19]. The escape of tumor cells from a primary aggressive tumor occurs as a series of steps beginning with local invasion and followed by the entry of tumor cells into blood vessels (intravasation), the survival of cancer cells in blood circulation, and the outflow of blood vessels (extravasation) in a specialized microenvironment, which serves as a previously prepared niche for secondary tumor growth (Figure 1A) [20]. Therefore, the metastatic process is not a disorderly process; on the contrary, it takes place as a series of well-coordinated cellular and molecular events driven by dynamic, genetic, and epigenetic programs. At the cellular level, alterations in the tight contacts between cancer cells and extracellular matrix (ECM) proteins occur, characterized by a morphology change known as an epithelial–mesenchymal transition (EMT), which allows for cancer cells to acquire enhanced migratory abilities, to detach from the ECM, and to initiate the invasion of proximal tissues. Diverse specialized transmembrane proteins drive the EMT, and alterations in their functions may result in chemo/radiotherapy resistance [21]. For instance, E-cadherin is a cell adhesion protein that keeps cells together and whose low expression is related to EMT activation, whereas high expression of its family member N-cadherin is associated with the loss of cell adhesions mediated by integrins, which are mechano-transduction proteins that sense the ECM (Figure 1B). During the invasion of nearby tissues, the ECM is degraded by the actions of metalloproteinases (MMPs) and the urokinase plasminogen activator (uPA) system [22]. Then, coordinated tumor cell migration is carried out with additional cellular processes, including the activation of the “angiogenic switch”, characterized by a change in the local balance of proangiogenic factors, such as the vascular endothelial growth factor (VEGF), and antiangiogenic factors, including thrombospondins and endostatins, among others, which indicates that angiogenesis is a key mechanism for tumor growth and invasive behavior [22]. Interestingly, more recent studies have been focused on the identification of potential regulatory non-coding RNAs such as microRNAs, long non-coding RNAs, and circular RNAs (circRNAs) as molecular drivers of metastasis and therapy resistance, which may lead to the identification of potential biomarkers and novel therapeutic targets [23,24].

## 3. Circular RNAs: New Players in Gene Expression

In 1976, circRNAs were observed in the light of confocal microscopy, by Sanger et al., for the first time. At first, these sequences were thought to be viroids due to their circular shape. Over time and with technological advances, they went from being classified as rare types of RNA that resulted from splicing artifacts or genetic rearrangements to real regulatory elements in eukaryotic cells [25]. CircRNAs are derived from precursor-messenger RNAs (pre-mRNAs), which are transcribed by RNA polymerase II and characterized by circular shapes that result from continuous, covalently closed single-stranded loops. These unique molecular structures make circRNAs resistant to RNAse-R and exonuclease digestion; hence, they exhibit high RNA stability [26]. CircRNAs are classified into three types, as follows, according to their structures and mechanisms of circularization: (i) exon circRNAs (ecircRNAs), (ii) circular intron RNAs (ciRNAs), and (iii) exon–intron circRNAs (ElciRNAs) [26]. Consisting of a single exon or multiple numbers of exons and formed through a shearing process called “head-to-tail” or “backsplicing,” ecircRNAs make up more than 80% of circRNAs and mostly exist in the cytoplasm. EIciRNAs are predominantly located in the nucleus and circularized in the form of retention introns between exons [27]. There are currently three models of circular RNAs production, namely, (i) intron-pairing-driven circularization, (ii) RNA-binding protein (RBP)-dependent circularization, and (iii) loop-driven circularization, which have been recognized to produce EcircRNAs and EIciRNAs (Figure 2A) [24]. Recent studies have found that circRNAs are exported to cytoplasm through a mechanism that is dependent on the length of a closed RNA molecule. The UAP56/DDX39B complex is required for the export of long circRNAs, whereas the URH49/DDX39A proteins are required for the transport of short circRNAs (<400 nt). Interestingly, the depletion of the UAP56 or URH49 factors can affect stationary levels of cytoplasmic circRNAs (Figure 2B). Interestingly, circRNAs that carry introns have been found to be retained in the nucleus and regulate the expression of their parental genes [28,29,30]. Recent investigations have demonstrated that circRNAs can also be exported from cancer cells as the cargo of lipidic extracellular vesicles (EVs) and can thus be detected in the circulation and urine [31].

## 4. Roles of Circular RNAs in Cancer Cells

### Functions in RNA Polymerase II Transcription and Splicing

A better-described function of circRNAs is that of being microRNA sponges because they can bind to those tiny regulatory ncRNAs; thus, they are also dubbed endogenous competitors [31]. CircRNAs have both oncogenic and tumor-suppressing properties in human cancer depending on the functions of the kidnapped microRNA (Figure 3A). Interestingly, some circRNAs have been discovered to also exert functions in transcriptional regulation through interaction with specific DNA loci to form RNA–DNA hybrids, which will cause a transcriptional pause and the recruitment of the splicing factors, as in the cases of circSEP3 and circSMARCA5 [32,33]. RBPs are important to the regulation of gene expression and splicing. Notably, circEIF3J and circPAIP2 have been detected to associate with the RNA polymerase II protein and enhance gene expression. Another example is intronic ci-ankrd52, which upregulates transcriptions driven by RNA polymerase II [34]. Likewise, the EIF3J and PAIP2 circRNAs interact with U1 snRNP, which, in turn, interacts with RNA polymerase II in the promoter regions of their parental genes, increasing transcriptions (Figure 3C) [35]. Another example is circ-UBR5, which regulates RNA splicing when it binds to regulators like QKI, NOVA1, and the U1 proteins (Figure 3B) [36]. CircRNAs also may contain small open-reading frames (ORFs) to encode polypeptides that mediate diverse cellular activities. An example is circ-ZNF609, which binds to polysomes in murine and human myoblasts to produce a 30 kDa protein in a splicing-dependent and 5´cap-independent fashion (Figure 3D) [37].

## 5. Circular RNAs Functions in Triple-Negative Breast Cancer

### CircRNAs Sponge Relevant MicroRNAs Involved in Tumor Progression and Metastasis

At least 60% of the human transcriptome belongs to ncRNAs that participate in biological processes as differential regulators with pathological impacts [38]. The role of circRNAs in breast cancer, especially in the triple-negative subtype, has recently been demonstrated. Some studies have shown that circRNAs act mainly through the kidnapping of microRNAs, which regulates important oncogenic and tumor-suppressor genes involved in the regulation of cancer hallmarks. Moreover, breast cancer metastasis has been associated with alterations in the expressions of multiple circRNAs and microRNAs (Table 1). For instance, Xu et al. demonstrated that the circular RNA known as circIKBKB activates the NF-κB pathway by promoting the IKKβ-mediated phosphorylation of IκBα, resulting in the inhibition of the IκBα feedback loop and facilitating the binding of NF-κB to the promoters of multiple bone-remodeling genes, stimulating breast cancer metastasis to bone [39]. On the other hand, circBCBM1 was detected in the brain as a promoter of metastasis through the modulation of the miR-125a/BRD4 axis [40]. Another relevant circular RNA is circZEB1, which has been markedly overexpressed in TNBC tissues and cell lines, promoting proliferation, and reducing apoptosis through the miR-448/eEF2K axis [41]. Notably, other circRNAs, such as circSKA3 and circRNA-UCK2, have been associated with tumor progression in TNBC [42,43], whereas the high expression of circPDCD11 has been closely correlated with an unfavorable prognosis in breast cancer patients [44]. Finally, circ-UBAP2, circGFRA1, and circEPSTI1 have also been related to TNBC progression [45,46,47]. Chen et al. showed that circHIF1A is involved in TNBC progression, since it was detected in breast cancer tissues with lymph node metastases [48]. The FUS (fused in Sarcoma) protein participates in the biogenesis of circHIF1A, which, in turn, binds to miR-149-5p. The FUS protein is transcriptionally activated by the NFIB protein, which can activate the AKT/STAT3 signaling pathways and the inhibition of the p21 protein, regulating cell proliferation and forming a circHIF1A/NFIB/FUS positive feedback loop. Interestingly, Chen et al. proposed that circHIF1A could be packaged and exported in exosomes [48].

Similarly, the overexpression of circBACH2 has been reported to facilitate the EMT, activating cancer cell invasion and migration as well as proliferation through the binding of both tumor suppressors miR-186-5p and miR-548c-3p, thus promoting the expression of the oncogenic chemokine receptor CXC type 4, or CXCR4. Moreover, circBACH2 was found expressed on the surfaces of various types of cancer cells; thus, its depletion has resulted in the suppression of the malignant progression of TNBC cells [50]. Another interesting study has demonstrated that circSEPT9 is overexpressed in TNBC tissues and its high expression levels correlate with advanced clinical stages and poor prognosis. Mechanistically, circSEPT9 could regulate the expression of the leukemia inhibitory factor (LIF) by sponging miR-637 and thus activating the LIF/Stat3 signaling pathway, which is involved in the progression of TNBC [51].

In another study conducted on normal human epithelial cells and TNBC cell lines, the overexpression of circANKS1B was associated with invasion and metastasis of breast cancer cells through the activation of the EMT. The identified targets of the circANKS1B were miR-148a-3p and miR-152-3p, which increased the expression of the transcription factor USF1, which could transcriptionally regulate the expression of TGF-β1 and activate the TGF-β1/Smad signaling pathway, promoting the EMT in this way [67]. Yang et al. detected the overexpression of another circular RNA, dubbed circAFG1, which binds to miR-195-5p, whose transcript target is the cyclin E1 (CCNE1) cell cycle protein. Functional analysis has demonstrated that the circAGFG1/miR-195-5p/CCNE1 axis promotes TNBC progression [52].

Intriguingly, Li et al. discovered that a circRNA derived from the HER2 gene, circ-HER2, was expressed in TNBC even when its classification indicated the absence of the HER2 receptor. The expression of the circ-HER2 contributed to homo/hetero EGFR/HER3 dimerization, which originated the sustained phosphorylation of AKT that promotes proliferation, invasion, and metastasis. The circ-HER2 expression correlated with poor prognosis in patients [53]. Other researchers have found other overexpressed circRNAs related to cell proliferation, invasion, and metastasis in TNBC: for instance, circRPPH1, which regulates the miR-556-5p/YAP1 axis. The YAP protein is an important transcriptional coactivator and regulates the Hippo-associated signaling transduction pathways in breast cancer [54]. On the other hand, the overexpression of hsa_circ_0131242 has been consistent with advanced stages of TNBC (II-III TNM) and larger tumors [55]. The hsa_circ_0000520/miR-1296 axis has stimulated the expression of the Zinc Finger X-chromosomal (ZFX) oncoprotein involved in the regulation of transcription in TNBC [56]. Likewise, circGNB1 regulates tumor progression through the regulation of the miR-141-5p/IGF1R axis and has been involved in poor prognosis [57]. Similarly, the downregulation of circUSP42 and the overexpression of its target, miR-182, have been associated with lymph node metastasis and poor prognosis [58].

Remarkably, circRNAs have been described to not only promote the metastasis of cancer cells but possibly also play the opposite role: strong tumor suppression. For example, the overexpression of circCDYL has promoted apoptosis and inhibited cell proliferation through regulation of the miR-190a-3p/TP53INP1 axis, therefore upregulating the tumor-suppressing TP53INP1 protein in TNBC [59]. Another relevant circRNA with tumor-suppressing functions is circNR3C2 (has_circ_0071127), which has been repressed in TNBC and negatively correlated with metastasis and cancer lethality [60]. Similarly, Xiao et al. demonstrated the antitumor role of circAHNAK1, which negatively regulates the miR-421/RASA1 axis. RASA1, a GTPase protein involved in the control of cell proliferation and differentiation, was suppressed by increasing levels of circAHNAK1 [61].

Finally, circRNA molecules could be derived from diverse genomic regions, as in the case of hsa_circ_0091074, which is generated from a genomic region of the X-specific inactive transcript (XIST) and binds to miR-1297, partially blocking the antitumor action of its target microRNA and resulting in the overexpression of the TAZ protein and positive regulation of the cell cycle in TNBC cells [62].

## 6. Circular RNAs Exported in Exosomes in TNBC

In recent decades, exosomes have become an important topic of study given their important roles in the regulation of cancer hallmarks. Exosomes are nanometric vesicles (40–100 nm in diameter) of endocytic origin that share a similar topology and lipid composition with the plasma membrane. From inside, a wide variety of cargo molecules, such as nucleic acids, proteins, and enzymes capable of modulating cellular activities in recipient cells through the transfer of functional genetic information, can be found [68]. Exosome biogenesis is mediated by two pathways that classify the proteins that they contain. The endosomal sorting complex transport (ESCRT)-dependent pathway is the best-characterized mechanism, involving four large protein complexes and more than 30 proteins. The ESCRT-independent pathway involves the inhibition of a neutral sphingomyelinase (nSMase) that is necessary to hydrolyze sphingomyelin and originate ceramide [69]. Exosomes are formed by budding from the membranes of multivesicular bodies (MVBs) from late endosomes: invaginations, called intraluminal vesicles (ILVs), that contain cytosolic components. MVBs can fuse with lysosomes for their degradation or may follow the endocytic pathway for the generation of exosomes [70].

Exosomal circRNAs are known to play an important role in cancer biology, as they can be taken up by neighboring or distant cells and affect many aspects of the physiological and pathological conditions of recipient cells. Recently, exosomes were discovered to be enriched with circRNA molecules, as demonstrated by Li et al. More than 1000 circRNAs have been identified in human serum exosomes. Notably, the circRNA content in exosomes is regulated by changes in the levels of the associated microRNAs in the producer cells [71]. On the other hand, Yang et al. found that circPSMA1 was overexpressed in serum exosomes from TNBC patients. Mechanistically, the circPSMA1 kidnapped miR-637, reducing cell proliferation, migration, and metastasis through AKT1 and β-catenin signaling in triple-negative MDA-MB-231 breast cancer cells [72].

An exosome-expression-profiling study performed on breast cancer cells and the serums of patients with aggressive metastatic tumors revealed that 1137 circRNAs were differentially expressed in exosomes from MDA-MB-231 cells. In addition, 1147 and 1195 circRNAs were deregulated in the exosomes of patients with metastatic and well-localized tumors, respectively. Furthermore, hsa_circ_0009634, hsa_circ_0020707, hsa_circ_0064923, hsa_circ_0104852, and hsa_circ_0087064 were found to have significant differential expressions in the exosomes of both cell lines and clinical tumors [49]. Another recent study showed the upregulation of circHIF1A in the exosomes of hypoxic cancer-associated fibroblasts (CAFs). Remarkably, circHIF1A from CAF-derived exosomes, transported into MDA-MB-231 cancer cells, has induced the modulation of the miR-580-5p/CD44 axis and its stemness properties [73].

## 7. Circular RNAs Regulation of Resistance to Chemo/Radiotherapy

The expression of circRNAs related to the doxorubicin (DOX) response has been reported in TNBC. DOX is a cytotoxic anthracycline antibiotic that commonly generates resistance [74]. The high expression of the circular RNA dubbed circUBE2D2 has been associated with a poor prognosis of TNBC patients in the advanced stages of the disease, as well as with lymph node metastases and resistance to DOX. CircUBE2D2 acts as a molecular sponge for miR-512-3p, which has a role as a suppressor of various tumors by acting on cell division cycle associated protein 3 (CDCA3) [63]. On the other hand, the hsa_circ_0092276/miR-384/ATG7 axis promotes autophagy and DOX resistance through the overexpression of hsa_circ_0092276 [64].

Remarkably, circRNAs that perform the reverse function have been studied. For instance, the exogenous overexpression of circKDM4C has inhibited DOX resistance as well as proliferation and metastasis in breast cancer. CircKDM4C is a sponge for miR-548p, and the overexpression of this miRNA reverses the attenuation of malignant phenotypes [65].

Xu et al. proposed circSMARCA5 as a potential biomarker to prevent resistance to anticancer drugs since the low expression of this circular RNA in breast cancer tissues has reduced sensitivity to cytotoxic drugs. This was achieved because circSMARCA5 binds to its parent gene locus (SMARCA5) and forms a DNA R-loop, causing a transcriptional pause in exon 15 of SMARCA5. This protein participates in chromatin remodeling in DNA-damaged regions and promotes the recruitment of factors involved in DNA repair [66].

Several studies have corroborated the participation of circRNAs in the modulation of the proapoptotic and antiapoptotic proteins involved in resistance to chemotherapy. The overexpression of circAMOTL1 in TNBC has been related to significant increases in paclitaxel resistance [75]. The role of circRNAs in apoptosis reduction, that is, modulating the expressions of AKT and proapoptotic factors BAX and BAK as well as the antiapoptotic BCL-2 protein, has been well-documented [76]. Another study has shown that hsa_circ_0000199 sponges both miR-613 and miR-206, which results in the activation of the PI3K/AKT/mTOR pathway, but when the hsa_circ_0000199 was silenced, increases in the chemosensitivity of the TNBC cell lines to the drugs cisplatin, adriamycin, paclitaxel, and gemcitabine were observed [77]. On the other hand, Zang et al. identified that the circ-RNF111/miR-140-5p/E2F3 axis is related to resistance to paclitaxel through the positive regulation of transcription factor E2F3 [78].

Wang et al. analyzed another circular RNA involved in resistance to paclitaxel: circWAC, which sponges the miR-142 responsible for the modulation of WWP expression and the activity of the PI3K/AKT pathway. The overexpression of miR-142 has improved the prognosis of TNBC patients, reducing both cell proliferation and invasion [79]. On the other hand, some studies have shown that circ_0085495, circ_0001667, and circ_0006528 expression are decreased and associated with cell proliferation as well as resistance to ADM in these cases [80,81,82]. Likewise, Wu et al. demonstrated that the activation of the circ-MMP1/miR-153-3p/ANLN axis promotes the lapatinib resistance in breast cancer cells, which is mediated by the exosome transfer of circ-MMP11 [83]. The activation of the circFAT1/microRNA-525-5p/SKA1 axis regulates the chemoresistance to oxaliplatin as well as cell migration, invasion, and apoptosis by activating the Notch and Wnt signaling pathway [84]. On the other hand, Zhu et al. confirmed that circFBXL5 suppresses resistance to 5-fluorouracil (5-FU) in TNBC through the modulation of the miR-216b/HMGA2 axis [85].

CircRNAs have been found to also demonstrate activity that improves the chemoresistance of treatments against TNBC. Li et al. showed that the overexpression of hsa_circ_0025202 prevents cell proliferation, tumorigenesis, and resistance to tamoxifen via the miR-197-3p/HIPK3 regulatory axis [86].

Radiotherapy is a common treatment for TNBC patients. Some of its types that are applied to patients include three-dimensional conformal radiotherapy (3D-CRT), intensity modulated radiotherapy (IMRT), and volumetric modulated arc therapy (VMAT) [87]. Circular RNAs also participate in the responses to radiotherapy; for example, Song et al. investigated the role of the circ-ADAM9/miR-383-5p/PFN2 axis in the radiosensitivity and apoptosis of cancer cells. The silencing of circ-ADAM9 was found to inhibit cell proliferation, migration, and invasion, and increase the radiosensitivity of the breast cancer cells [88].

Another study has demonstrated changes in the cell viability and proliferation of radium-resistant breast cancer cells. For example, circ-ABCC1 was found to be upregulated in resistant cells compared to the corresponding parental cells. The forced inhibition of miR-627-5p, a target of circ-ABCC1, resulted in the suppression of radioresistance [89]. The explanation of tumor heterogeneity by the variable sensitivity and unequal response of tumor cells to radiotherapy has been considered. Tumor recurrence has been related to the presence of self-renewing stem-like cancer cells [90]. Tumor cells can acquire stem cell characteristics to repopulate the tumor mass after radiation treatment [91]. In a recent study, researchers subjected MCF7 breast cancer cells to fractional doses of X-rays and found a proinflammatory immune response like that of cancer stem cells and the activation of procancer survival-signaling pathways related to radioresistance [92]. Estrogen signaling was also downregulated; it possibly promotes cross-resistance to tamoxifen and endocrine therapies. Kong et al. demonstrated that eliminating hsa_circ_0008500 improved the radiosensitivity of breast cancer and suppressed tumor growth in vivo [93]. Finally, the overexpression of circNCOR1 favors cell proliferation in MDA-MB-231 and BT549 breast cancer cells, but radiosensitivity has been affected by the regulation of CDK2 via miR-638 interactions [94].

## 8. Conclusions

CircRNAs are a type of ncRNA that has various regulatory roles in aggressive TNBC. Evidence has shown that they may greatly influence cell proliferation, migration, metastasis, apoptosis, and chemoresistance. CircRNAs have a dual role in metastasis, as they may act as oncogenes and tumor-suppressing genes through the regulation of microRNAs. Furthermore, circRNAs have been detected in exosomes in the serums of TNBC patients, representing potential biomarkers of disease, and participate in the response to anticancer drugs in TNBC. The relevance of this review offers an in-depth overview of the impacts of different circRNAs on TNBC biology and highlights their potential as novel therapeutic targets.

## Figures and Tables

**Figure 1 ncrna-09-00055-f001:**
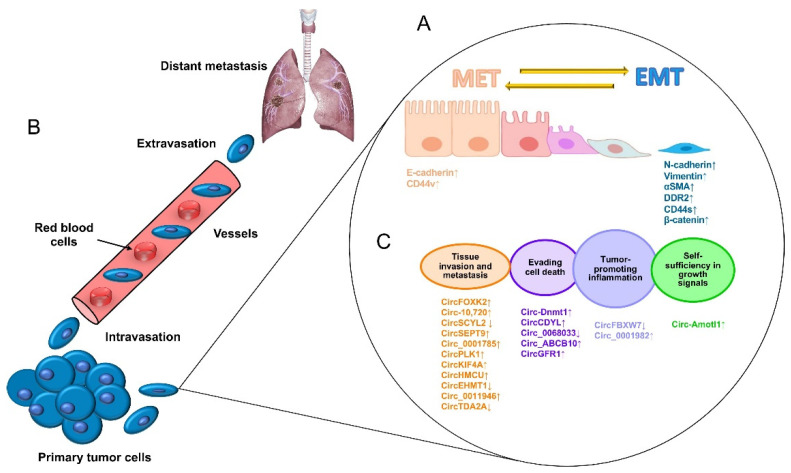
The process of the metastasis of tumor cells. (**A**) Metastasis begins with paracrine signals that activate genes to initiate the switch from the mesenchymal–epithelial transition (MET) to the EMT status, changing the interactions of cancer cell junctions and extracellular matrix proteins, including cadherins, vimentin, CD44, and beta-catenin, among others. (**B**) Next, cancer cells break off from the primary tumor and move through the wall of blood vessels (intravasation) to travel through the bloodstream in search of a new site to grow (extravasation). The cells leave the bloodstream to settle in tissues surrounding the blood vessels from which they emerged. (**C**) Diverse circular RNAs have been identified as regulators of genes that participate in cell proliferation, apoptosis inflammation, apoptosis, tissue invasion, and metastasis. ↓ Downregulated. ↑ Upregulated.

**Figure 2 ncrna-09-00055-f002:**
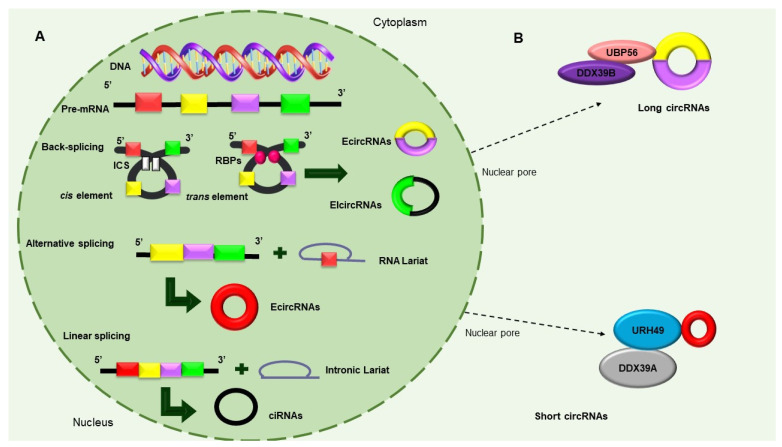
Biosynthesis of circular RNAs. (**A**) Precursor mRNA (pre-mRNA) consensus sequences, such as the cis elements known as intronic complementary sequences (ICSs) and flanking circRNA-forming exons, in addition to trans elements, including some RBPs, participate in the formation of circRNAs. The backsplicing process originates EcircRNAs and circular RNAs with introns (EIcircRNAs) via the donor splicer and the receptor-splicing site through base-pairing interactions between the ICSs or the dimerization of the RBPs. Alternative splicing circularization allows the covalent attachment of an upstream exon and a downstream exon that generates a lariat RNA precursor. When an intronic lariat is formed via the canonical splicing pathway, it generates a linear lariat intron that contains skipped exons that eventually undergo reverse splicing. (**B**) The export of circRNAs from the nucleus to the cytoplasm is dependent on their size. The UBP56 protein is responsible for the export of long circRNAs (>1300 nucleotides), while URH49 exports short circRNAs (<400 nt) [28,29,30].

**Figure 3 ncrna-09-00055-f003:**
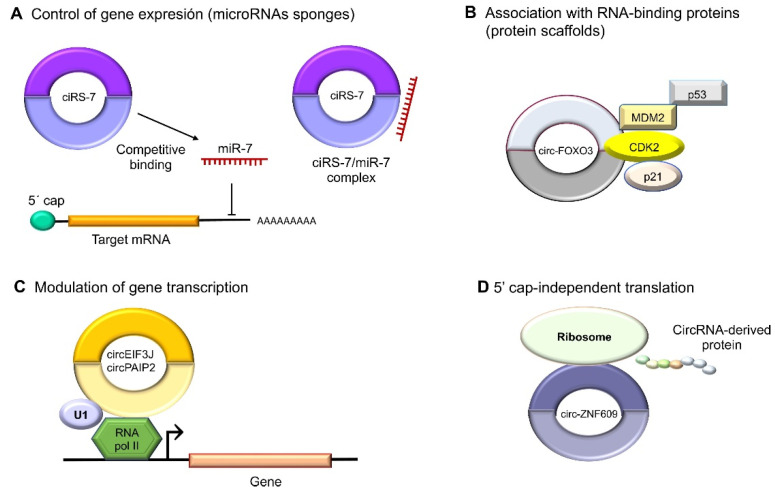
Molecular functions of circular RNAs. (**A**) CircRNAs act indirectly on gene expression as microRNA sponges: for example, oncogenic ciRS-7 (CDR1as) competitively binds to tumor suppressor miR-7, since it contains miRNA response elements (MREs). (**B**) CircRNAs can also bind to RBPs such as circ-FOXO3, which interacts with MDM2 (murine double-minute 2) to degrade p53 by ubiquitination, forming a ternary complex with CDK2 and p21. (**C**) The nuclear circRNAs that conserve the intronic sequences of their parental genes, circEIF3J and circPAIP2, to improve those genes’ cis expressions could associate with RNA polymerase II, which would improve their functions in a U1 snRNP-dependent manner. (**D**) Multiple circRNAs contain internal ribosome entry sites (IRESs), which suggests that they can be translated into peptides or proteins in a 5′ cap-independent manner, as in the case of circ-ZNF609, the translation of which plays an important role in myogenesis.

**Table 1 ncrna-09-00055-t001:** Circular RNAs modulated in TNBC and the regulatory axes they regulate.

CircRNA	Expression in TNBC	Regulated Axis	Functions and Prognosis	Reference
circZEB1	↑	circZEB1/miR-448/eEF2K	Cell proliferation.	[41]
circSKA3	↑	-	Tumor progression. Poor prognosis. Exosomal transmission.	[42]
circ-PDCD11	↑	circ-PDCD11/miR-512-3p/CDCA3	Accelerated glucose uptake, lactate production, ATP generation, and extracellular acidification. Poor prognosis of TNBC.	[44]
circRNA-UCK2	↑	circRNA-UCK2/miR-767-5p/TET1	Tumor progression. Poor prognosis.	[43]
circANKS1B	↑	circANKS1B/miR-148a-3p/miR-152-3p/USF1	Metastasis.	[45]
circGFRA1		circGFRA1/miR-361-5p/TLR4	Resistance to paclitaxel.	[46]
circEPSTI1	↑	circEPSTI1/miR-4753/BCL11A	Cell proliferation.	[47]
circHIF1A	↑	circHIF1A//NFIB/FUScircHIF1A/miR-580-5p/CD44	Packaged into exosomes. Metastasis and poor prognosis of TNBC.	[48,49]
circBACH2	↑	circBACH2/miR-186-5p/miR-548c-3p/CXCR4	Cell proliferation, invasion, and metastasis.	[50]
circSEPT9	↑	circSEPT9/miR-637/LIF	Facilitation of the carcinogenesis and development of TNBC.	[51]
circAGFG1	↑	circAGFG1/miR-195-5p/CCNE1	Cancer progression.	[52]
circ-HER2	↑	-	Cell proliferation, invasion, and metastasis.	[53]
circRPPH1	↑	circRPPH1/miR-556-5p/YAP1	Cell proliferation, invasion, and metastasis.	[54]
hsa_circ_0131242	↑	has_circ_0131242/miR-2682	Cancer progression.	[55]
hsa_circ_0000520	↑	hsa_circ_0000520/miR-1296/ZFX	Regulation of transcription.	[56]
circGNB1	↓	circGNB1/miR-141-5p/IGF1R	Regulation of tumor progression.	[57]
circUSP42	↑	circUBE2D2/miR-512-3p/CDCA3	Cell proliferation, metastasis, and chemoresistance.	[58]
circCDYL	↓	circCDYL/miR-190a-3p/TP53INP1	Tumor suppression.	[59]
circNR3C2	↓	circNR3C2/miR-513a-3p/HRD1	Inhibition of cell proliferation, migration, invasion, and EMT process.	[60]
circAHNAK1	↑	circAHNAK1/miR-421/RASA1	Cell proliferation.	[61]
hsa_circ_0091074	↑	hsa_circ_0091074/miR-1297/TAZ	Regulation of cell cycle.	[62]
circUBE2D2	↑	circUBE2D2/miR-512-3p/CDCA3	Cell proliferation, metastasis, and chemoresistance.	[63]
hsa_circ_0092276		hsa_circ_0092276/miR-384/ATG7	Autophagy and DOX resistance.	[64]
CircKDM4C	↓	CircKDM4C/miR-548p	Inhibited DOX resistance.	[65]
circSMARCA5	↓	circSMARCA5/miR-548p/SMARCA5	Prevention of chemoresistance.	[66]

↑: upregulated expression. ↓: downregulated expression.

## Data Availability

Not applicable.

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
