# Peer review of "Novel Insights into Circular RNAs in Metastasis in Breast Cancer: An Update"

_ncrna, 2023, doi:10.3390/ncrna9050055_

Round 1
Reviewer 1 Report
This manuscript by Zepeda-Enríquez et. al. sets out to review repertoire of circular RNAs in metastatic breast cancer. Circular RNAs represent a new class of noncoding RNAs whch are key regulators of breast carcinogenesis and progression. Overall, this is an interesting and well written review that described a comprehensive landscape of circRNAs in breast cancer research. However, there are several grammatical and syntax errors as well as some sections are less clear, so authors should fix those before the manuscript is accepted for publication. Following are my specific comments:
1. Introduction: “Breast cancer is an increasingly disease…” (line 25). What does it mean? This entire section (lines 25-27) dealing with cancer and death numbers/stats has to be rewritten for clarity.
2. Introduction: Although the major theme of this manuscript is on the role of Circular RNAs (circRNAs), there is no mention of these RNAs in the Introduction section. Why?
3. Metastasis in breast cancer: “ Interestingly, more recent studies have been focused in the identification of potential regulatory non-coding RNAs such as microRNAs....”. Provide references/citations.
4. Figure 1: What is MET? If it is an abbreviation of metastasis, then it should be mentioned in parenthesis in the figure legend.
5. Circular RNAs: “Circular RNAs (circRNAs) are cycled endogenous regulatory non-coding RNAs…”. What “cycled” means here? Is it “cycled” or “circular”?
6. The classification of circular RNAs outlined in the manuscript (lines 119-122) is based on the genomic locations in the context of splice junction from which circRNAs originate. So authors should provide this contextual information and cite references appropriately.
7. Figure 2: Fix typo – “ex-port”. Cite UBP56 reference in Fig 2B
8. Section 4.1: “The better described function of circRNAs is as microRNAs molecular sponges…” cite references.
9. Section 5.1: “ For example, Chen et al. showed that 218 circHIF1A was involved…”. Cite reference.
10. Section 5: Given that there is a growing list of circRNAs in triple negative breast cancer, it might be a good idea to list these circRNAs in a tabular format alongside their putative function.
The paper is well written but it still needs some English improvements.
Author Response
Reviewer 1
This manuscript by Zepeda-Enríquez et. al. sets out to review repertoire of circular RNAs in metastatic breast cancer. Circular RNAs represent a new class of noncoding RNAs whch are key regulators of breast carcinogenesis and progression. Overall, this is an interesting and well written review that described a comprehensive landscape of circRNAs in breast cancer research. However, there are several grammatical and syntax errors as well as some sections are less clear, so authors should fix those before the manuscript is accepted for publication. Following are my specific comments:
Reply: We acknowledge to reviewer for the opportunity to revise and correct our manuscript. Your critical suggestions that we have fully replied greatly increase the quality of our study. All amendments have been marked in yellow color in the revised text for your easy reference and reading. We have carefully corrected the manuscript according to the referee suggestions and provide a point-by-point response.
- Introduction: “Breast cancer is an increasingly disease…” (line 25). What does it mean? This entire section (lines 25-27) dealing with cancer and death numbers/stats has to be rewritten for clarity.
Reply: Thanks for the comments. We apologize for the confusion in the description of breast cancer statistics. As you suggested, we have clarified and shortened the information in the introduction section (Page 1, lanes 25-27).
- Introduction: Although the major theme of this manuscript is on the role of Circular RNAs (circRNAs), there is no mention of these RNAs in the Introduction section. Why?
Reply: We appreciate the reviewers’ wise comments. Now, we have added literature about circular RNAs in introduction section (Page 2, lanes 62-70).
- Metastasis in breast cancer: “Interestingly, more recent studies have been focused in the identification of potential regulatory non-coding RNAs such as microRNAs....”. Provide references/citations.
Reply: References 24, 24 have been added (Page 3, lane 106).
- Figure 1: What is MET? If it is an abbreviation of metastasis, then it should be mentioned in parenthesis in the figure legend.
Reply: We have added the missing abbreviatures as follows: mesenchymal epithelial-transition (MET) and epithelial-mesenchymal transition (EMT) in figure 1 legend (Page 3, lanes 109-110).
- Circular RNAs: “Circular RNAs (circRNAs) are cycled endogenous regulatory non-coding RNAs…”. What “cycled” means here? Is it “cycled” or “circular”?
Reply: We apologize for the error. We have deleted the word “cycled”.
- The classification of circular RNAs outlined in the manuscript (lines 119-122) is based on the genomic locations in the context of splice junction from which circRNAs originate. So authors should provide this contextual information and cite references appropriately.
Reply: Thanks for the comments, we have outlined the corresponding references 26 and 27 (Page 3, lanes 131 and 135).
- Figure 2: Fix typo – “ex-port”. Cite UBP56 reference in Fig 2B
Reply: Typo has been corrected. Also, the corresponding reference has been added (Page 4, lane 165).
- Section 4.1: “The better described function of circRNAs is as microRNAs molecular sponges…” cite references.
Reply: Reference 31-33 have been added (Page 5, lane 172 and 177).
- Section 5.1: “For example, Chen et al. showed that 218 circHIF1A was involved…”. Cite reference.
Reply: Reference 48 has been added (Page 6, lane 228).
- Section 5: Given that there is a growing list of circRNAs in triple negative breast cancer, it might be a good idea to list these circRNAs in a tabular format alongside their putative function.
Reply: Thanks for the comments. We have added a new Table 1 to summarizes the circRNAs functions in TNBC (Page 7).
Comments on the Quality of English Language
The paper is well written, but it still needs some English improvements.
Reviewer 2 Report
This review by Zepeda-Enríquez et al, makes a timely contribution to the triple negative breast cancer field with emphasis on metastasis and the role of circular RNAs. The manuscript covers a broad area in a lot of detail. The inclusion of immunotherapy is welcome.
Some issues however:
1. Perhaps circular RNA responsive to radiation therapy could also be included or cited.
2. The last sentence of the abstract should be clarified. It should ideally be accompanied ‘by’.
3. The English needs a bit of adjustment –increasingly disease: line 25; ‘the’: line 63; ‘the’: line 75; on the contrary; line 76; ‘the’ before cancer remove: line 76; in coordination: line 88; processes: line 110; generating: line 148; export: line 152; encodes: line 177; Another instead of other: line 232; cancers: line 337;
4. Provide a reference for line 28 as 685,000 deaths seem low for world wide.
5. Correct the abbreviation EMC should be ECM: line 79
6. Vimentin spelt incorrectly: Figure 1
7. Define the blue and red discs in Figure 1.
8. Extravasation needs to also be placed in brackets in the Figure legend.
9. Evadin cell death ?
10. Clarify as this looks like duplication – line 115.
11. Quaking – include the isoform number.
12. Definitions of the circular RNAs are a bit baffling eg. EI-circRNA or EI-ciRNA, circular circRNA, EcircRNA (line 129), EIcircRNA, check Figure 2 line 145. Keep consistent eg circPS-MA1 versus circPSMA1 (line 308 and line 309). Is the circRNA nomenclature correct for circRNAs 1147 and 1195 (line 315)?
13. Clarify how nuclear export mechanisms via depletion of UAP56 and URH49 causes circRNA to be in the nucleus.
14. Full definition of RNA binding proteins (RBPs) needs to be included line 144.
15. Should SD and SA be in the figure?
16. Reference needed for ciankrd52 – line 168.
17. Clarify line 171.
18. Figure 3 translation spelling and proteins in 5’cap-independent either remove or improve wording.
19. Define MDM2 line 187.
20. What are the other proteins (pale blue) in complex with U1 in Figure 3C?
21. circKBKB presumably should be IKBKB? Line 203
22. References needed for line 208 after axis, and line 212 after patients.
23. Do you need word ‘circRNAs’ in line 212 or ‘miRNA’ in line 308?
24. Replace modulates with modulation of – line 361.
25. Remove ‘in’ line 383.
26. Change ‘its’ to ‘their’ line 388.
The English is generally good. In my report I listed the corrections that can improve the language quality.
Author Response
Reviewer 2
This review by Zepeda-Enríquez et al, makes a timely contribution to the triple negative breast cancer field with emphasis on metastasis and the role of circular RNAs. The manuscript covers a broad area in a lot of detail. The inclusion of immunotherapy is welcome.
Reply: We acknowledge to reviewer for the opportunity to revise and correct our manuscript. Your critical suggestions that we have fully replied greatly increase the quality of our study. All amendments have been marked in yellow color in the revised text for your easy reference and reading. We have carefully corrected the manuscript according to the referee suggestions and provide a point-by-point response. No immunotherapy data on circular RNAs and triple negative breast cancer have been reported.
Some issues however:
- Perhaps circular RNA responsive to radiation therapy could also be included or cited.
Reply: Thanks for the comments. We apologize for the omission. We have added to the review a new paragraph about the role of circRNAs on radiotherapy response, as you suggested (Page 9-10, lanes 384-408).
- The last sentence of the abstract should be clarified. It should ideally be accompanied ‘by’.
Reply: Abstract have been corrected.
- The English needs a bit of adjustment –increasingly disease: line 25; ‘the’: line 63; ‘the’: line 75; on the contrary; line 76; ‘the’ before cancer remove: line 76; in coordination: line 88; processes: line 110; generating: line 148; export: line 152; encodes: line 177; Another instead of other: line 232; cancers: line 337;
Reply: Thanks for the comments. We have corrected the grammar as reviewer suggested.
- Provide a reference for line 28 as 685,000 deaths seem low for world wide.
Reply: The reference 1 was originally added. The number of deaths for breast cancer is correct according to reference 1.
- Correct the abbreviation EMC should be ECM: line 79
Reply: Thanks for the comments. We have corrected the abbreviature (Page 2, lane 88).
- Vimentin spelt incorrectly: Figure 1
Reply: We have corrected the misspelling in figure 1 (Page 3).
- Define the blue and red discs in Figure 1.
Reply: We have corrected the figure 1 and indicated the blue disc as primary tumor cells, and the red discs as red blood cells.
- Extravasation needs to also be placed in brackets in the Figure legend.
Reply: We have added brackets to extravasation in figure legend 1 (Page 3, lane 114).
- Evadin cell death?
Reply: Thanks for the comment. We have replaced “evadin” by “evading” in figure 1.
- Clarify as this looks like duplication – line 115.
Reply: We have deleted the duplicated sentence.
- Quaking – include the isoform number.
Reply: We apologize for the confusion. Now, we have replaced “isoform” for “types on RNA”.
- Definitions of the circular RNAs are a bit baffling eg. EI-circRNA or EI-ciRNA, circular circRNA, EcircRNA (line 129), EIcircRNA, check Figure 2 line 145. Keep consistent eg circPS-MA1 versus circPSMA1 (line 308 and line 309). Is the circRNA nomenclature correct for circRNAs 1147 and 1195 (line 315)?
Reply: We have corrected all the misspellings on circular RNAs nomenclature. The nomenclature used here is accord with reference 26 Li, Z et al. Exon-intron circular RNAs regulate transcription in the nucleus. Nat Struct Mol Biol. 2015,22(3),256-264).
- Clarify how nuclear export mechanisms via depletion of UAP56 and URH49 causes circRNA to be in the nucleus.
Reply: We apologize for the confusion in the description of nuclear export mechanism. We clarified the correct description based on the references 28 and 29.
- Full definition of RNA binding proteins (RBPs) needs to be included line 144.
Reply: We have added a definition of RBP (Page 4, lane 157).
- Should SD and SA be in the figure?
Reply: Thanks for the comments. We have corrected the info.
- Reference needed for ciankrd52 – line 168.
Reply: We have added the corresponding reference number 34 (Page 5, lane 181).
- Clarify line 171.
Reply: Sentence has been corrected.
- Figure 3 translation spelling and proteins in 5’cap-independent either remove or improve wording.
Reply: We have improved the wording as “5’ cap-independent translation” in the figure 3D, as you suggested.
- Define MDM2 line 187.
Reply: We have defined MDM2 gene as murine double-minute 2 (Page 5, lane 187).
- What are the other proteins (pale blue) in complex with U1 in Figure 3C?
Reply: Thanks for the comments, we apologize for the mistake. We have corrected the figure and indicated only one protein (U1) in figure 3.
- circKBKB presumably should be IKBKB? Line 203
Reply: We apologize for the confusion in the description. We have corrected the circular RNA nomenclature (circIKBKB).
- References needed for line 208 after axis, and line 212 after patients.
Reply. References have been added.
- Do you need word ‘circRNAs’ in line 212 or ‘miRNA’ in line 308?
Reply: Thanks for the observations. Both words have been deleted in the revised version of manuscript.
- Replace modulates with modulation of – line 361.
Reply: The “modulates” word have been replaced by “modulation of”, as suggested.
- Remove ‘in’ line 383.
Reply: “In” word have been deleted.
- Change ‘its’ to ‘their’ line 388.
Reply: Changes have been made as you suggested.